# Pretreatment with Zonisamide Mitigates Oxaliplatin-Induced Toxicity in Rat DRG Neurons and DRG Neuron–Schwann Cell Co-Cultures

**DOI:** 10.3390/ijms23179983

**Published:** 2022-09-01

**Authors:** Shizuka Takaku, Kazunori Sango

**Affiliations:** Diabetic Neuropathy Project, Tokyo Metropolitan Institute of Medical Science, Tokyo 156-8506, Japan

**Keywords:** zonisamide, oxaliplatin, dorsal root ganglion neurons, ND7/23 cells, IFRS1 Schwann cells, cell death, co-culture, axonal degeneration, demyelination, signaling pathways

## Abstract

Oxaliplatin (OHP) is a platinum-based agent that can cause peripheral neuropathy, an adverse effect in which the dorsal root ganglion (DRG) neurons are targeted. Zonisamide has exhibited neuroprotective activities toward adult rat DRG neurons in vitro and therefore, we aimed to assess its potential efficacy against OHP-induced neurotoxicity. Pretreatment with zonisamide (100 μM) alleviated the DRG neuronal death caused by OHP (75 μM) and the protective effects were attenuated by a co-incubation with 25 μM of the mitogen-activated protein kinase (MAPK; MEK/ERK) inhibitor, U0126, or the phosphatidyl inositol-3′-phosphate-kinase (PI3K) inhibitor, LY294002. Pretreatment with zonisamide also suppressed the OHP-induced p38 MAPK phosphorylation in lined DRG neurons, ND7/23, while the OHP-induced DRG neuronal death was alleviated by pretreatment with the p38 MAPK inhibitor, SB239063 (25 μM). Although zonisamide failed to protect the immortalized rat Schwann cells IFRS1 from OHP-induced cell death, it prevented neurite degeneration and demyelination-like changes, as well as the reduction of the serine/threonine-specific protein kinase (AKT) phosphorylation in DRG neuron–IFRS1 co-cultures exposed to OHP. Zonisamide’s neuroprotection against the OHP-induced peripheral sensory neuropathy is possibly mediated by a stimulation of the MEK/ERK and PI3K/AKT signaling pathways and suppression of the p38 MAPK pathway in DRG neurons. Future studies will allow us to solidify zonisamide as a promising remedy against the neurotoxic adverse effects of OHP.

## 1. Introduction

Drug repositioning or repurposing is a strategy allowing us to discover novel benefits of drugs that are currently applicable to patients, a strategy that may significantly reduce the cost and time required for the development of drugs as compared to conventional approaches [1]. Zonisamide, a benzisoxazole derivative, was initially developed as an anti-epileptic drug [2] and has been successfully repositioned for Parkinson’s disease (PD) [3,4]. Recent studies have shed light on the neurotrophic and neuroprotective properties of zonisamide in the peripheral nervous system (PNS). Yagi et al. [5] have found that zonisamide can promote neurite outgrowth from cultured motor neurons and can facilitate axonal regeneration following sciatic nerve injury in mice. In a recent study of ours [6], zonisamide enhanced neurite outgrowth from cultured adult rat dorsal root ganglion (DRG) sensory neurons through the stimulation of the phosphatidyl inositol-3′-phosphate-kinase/serine/threonine-specific protein kinase (PI3K/AKT) and the mitogen-activated protein kinase (MAPK) kinase/extracellular signal-regulated kinase (MEK/ERK) signaling pathways. In addition, its mitigating effects against neuropathic pain in rodent models have been well-documented [7,8]. These findings suggest its potential repositioning for the PNS lesions; however, the underlying mechanisms remain largely unknown.

Oxaliplatin (OHP), a platinum-based agent that inhibits DNA replication and transcription, has been extensively utilized in the treatment of colorectal cancer and other solid tumors; however, it frequently evokes acute and/or chronic neuropathies as dose-limiting adverse reactions [9]. It is recognized that DRG neurons are the most common target of OHP and other platinum-based compounds, and a considerable number of in vivo and in vitro studies have been conducted on the mechanisms of the OHP-induced DRG neuronal cell injury and death, e.g., nuclear and mitochondrial DNA damage [10], oxidative stress [11], and activation of the p38 MAPK signaling pathways [12]. Although the pathogenesis-based therapeutic approaches toward OHP-induced neuropathy are ongoing, no effective remedies have been developed, thus far, and the available prevention and treatment strategies are currently limited to a dose-modification of the administered OHP and the prescription of symptomatic treatment agents [13,14]. Based on the protective activities of zonisamide toward the DRG neurons described above [6], its potential efficacy against the OHP-induced neurotoxicity defines the aim of the present study. Our findings suggest that pretreatment with zonisamide can alleviate the OHP-induced toxicity against primary cultured and lined DRG neurons via suppressing the p38 MAPK pathway, and also by stimulating the MEK/ERK and PI3K/AKT pathways. Although zonisamide failed to protect lined Schwann cells from OHP-induced cell death, it did manage to prevent neurite degeneration and demyelination-like changes in DRG neuron–Schwann cell co-cultures exposed to OHP.

## 2. Results

### 2.1. Zonisamide Pretreatment Alleviates OHP-Induced Toxicity against Primary Cultured Adult Rat DRG Neurons

Before investigating the zonisamide activities, we confirmed the OHP-induced toxicity against DRG neurons obtained from 8 to 12-week-old female Wistar rats. The cells were incubated for 24 h in a serum-free medium with different concentrations (0 μM, 5 μM, 75 μM, and 100 μM) of OHP. The cell survival assays using trypan blue stain revealed that OHP diminished the average viability ratio in a dose-dependent manner (Figure 1a); a finding that is in agreement with those of the previous studies using embryonic and mature Sprague–Dawley rat DRG neurons [12,15]. Based on these findings and the clinical relevance of the zonisamide application (i.e., the concentration of zonisamide in the sera of epilepsy patients was 50–200 μM) [16], we examined the effects of a 100 μM zonisamide preincubation on OHP-induced (75 μM) neurotoxicity; the cells were incubated for 24 h in the presence or absence of zonisamide and were then maintained for 24 h in the presence or absence of OHP (hereafter termed as the “(Control)”, “(Zonisamide)”, “(OHP)”, and “(Zonisamide + OHP)” groups). The representative phase-contrast micrographs (Figure 1b) indicate that zonisamide preincubation improved the viability and neurite outgrowth of OHP-treated DRG neurons. The significantly higher viability ratio observed in the Zonisamide + OHP group (than that in the OHP group; Figure 1b) suggests that the zonisamide pretreatment can alleviate the OHP-induced toxicity.

### 2.2. Zonisamide Pretreatment Alleviates the OHP-Induced Toxicity against Immortalized Rat DRG Neurons, ND7/23, but Not against Immortalized Rat Schwann Cells, IFRS1

In addition to primary cultured DRG neurons, the OHP-induced toxicity against immortalized rat DRG neurons (ND7/23) [17,18] and Schwann cells (IFRS1) [19] was evaluated by employing an MTS assay. These lined cells have been utilized for the study of PNS lesions as substitutes for the primary cultured cells. In a similar manner to DRG neurons (Figure 1b), a pretreatment with zonisamide (100 μM) for 6 h was able to mitigate the ND7/23 cell death due to exposure to OHP (75 μM) for 24 h; the relative cell viability in the Zonisamide + OHP group was significantly higher than that in the OHP group (Figure 2a). The relative IFRS1 cell viability was significantly lower in the OHP group than in the control group, but the degree of reduction was more subtle compared to DRG neurons and ND7/23 cells (Figure 2b). The prolongation of the OHP exposure time from 24 h to 48 h resulted in more detrimental IFRS1 cell loss (Figure 2c); however, no significant differences in terms of the cell viability between the OHP and the Zonisamide + OHP groups were observed regardless of the addition of the duration of OHP exposure (Figure 2b,c). In addition to IFRS1 cells, zonisamide pretreatment failed to restore the OHP-induced cell death of immortalized mouse Schwann cells 1970C3 [20] (Appendix A). These findings suggest no protective activity of zonisamide toward Schwann cells regardless of cell lines and agree with its lack of any stimulating effects on the proliferation/survival or migration of IFRS1 cells in our previous study [6].

### 2.3. Involvement of p38 MAPK and MEK/ERK Signaling Pathways in Zonisamide’s Alleviating Effects on the OHP-Induced DRG Neuronal Death

Several studies have indicated the involvement of the MAPK signaling pathways (p38 MAPK, MEK/ERK, and JNK/SapK) in the OHP-induced DRG neuronal death [12,21]. By employing Western blotting analysis, we observed an enhanced p38 MAPK phosphorylation and a reduced ERK phosphorylation in ND7/23 cells exposed to OHP, and these changes were attenuated by pretreatment with zonisamide. Neither OHP nor zonisamide affected the JNK phosphorylation (Figure 3a). Based on these findings and the zonisamide-induced ERK phosphorylation in ND7/23 cells [6], we further investigated the involvement of the p38 MAPK and MEK/ERK signaling pathways in the assessed zonisamide actions by using inhibitors of the respective pathways. Pretreatment with 25 μM of SB239063, a p38 MAPK inhibitor, exerted the alleviating activity against OHP to an extent nearly comparable to that of zonisamide (Figure 3b), whereas a co-incubation with 25 μM of U0126, a MEK inhibitor, tended to weaken the alleviating activity of zonisamide (Figure 3c). The latter finding is in agreement with the attenuating effects of U0126 against zonisamide-induced DRG neurite outgrowth [6]. Taking these findings together, zonisamide’s alleviating effects on the OHP-induced DRG neuronal death can be attributable to the suppression of the p38 MAPK pathway and the restoration of the MEK/ERK pathway.

### 2.4. Involvement of the PI3K/AKT Signaling Pathway in Zonisamide’s Alleviating Effects on the OHP-Induced DRG Neuron Death

Besides the MAPK signaling pathways, our previous study suggests that the PI3K/AKT signaling pathway plays a role in the stimulating effects of zonisamide on the neurite outgrowth of DRG neurons [6]. In the present study, the pretreatment with zonisamide tended to upregulate the expression of phospho-AKT (when compared with the (Control) group) but did not affect its expression in OHP-treated ND7/23 cells (Figure 4a). In contrast to these findings, the co-incubation with 25 μM of LY294002, a PI3K inhibitor, canceled the protective activity of zonisamide against the OHP-induced DRG neuronal death (Figure 4b); This finding agrees with the inhibitory effects of LY294002 against zonisamide-induced DRG neurite outgrowth [6] and suggests that zonisamide’s alleviating effects on the OHP-induced DRG neuronal death can be attributable to the restoration of the PI3K/AKT pathway.

### 2.5. Pretreatment with Zonisamide Alleviates the OHP-Induced Neurite Degeneration and Demyelination-like Changes in DRG Neuron–IFRS1 Co-Cultures

IFRS1 cells are recognized as one of the few lined Schwann cells that possess the capability to myelinate neurites in co-culture with neurons [19]. The co-culture of DRG neurons and IFRS1 Schwann cells was maintained for 14 days under a serum-free culture condition in the presence of ascorbic acid (50 μg/mL) to induce myelination, as previously described [22]. The cells were then incubated for 2 days in the presence or absence of 100 μM of zonisamide and then were subsequently maintained for 7 days in the presence of OHP (10 μM; Figure 5a). As the OHP at 75 μM was too toxic for the co-culture, the reduced concentration (10 μM) was applied to allow us to sustain the tissue structure. By observation under a phase-contrast microscope, the pretreatment with zonisamide alleviated the OHP-induced neurite degeneration, as well as the detachment of IFRS1 cells from the neurite networks in the co-culture (Figure 5b). We subsequently undertook immunocytochemistry and Western blotting by using the co-cultured cells. However, the former did not work because the OHP-treated samples were fragile and would easily detach from the dishes during the fixation and washing. Western blotting analysis revealed that the pretreatment with zonisamide was able to restore the downregulated expression of phospho-AKT in the OHP-treated cells (Figure 5c).

## 3. Discussion

Zonisamide has been shown to alleviate neuropathic pain, one of the most characteristic symptoms of peripheral neuropathies, via blocking sodium and calcium channels [23] and by suppressing microglial activation [8]. In addition to its efficacy as a symptomatic treatment agent, recent studies by us [6] and others [5,24] have demonstrated zonisamide’s protective activity toward the PNS and have implied its potency as a pathogenesis-based remedy against peripheral neuropathies. In the present study, pretreatment with zonisamide was able to significantly alleviate the OHP-induced cell death of both primary cultured and lined (ND7/23) DRG neurons. The zonisamide concentration used in this study (100 μM) is within the range of concentrations that can be found in the sera of epilepsy patients (50–200 μM) [16]. Likewise, the OHP concentration at 75 μM falls also within the clinical concentration range (40–500 μM) [25]. Therefore, it seems fair to suggest that our research protocols have some clinical relevance. The ND7/23 cells, deriving from a hybridoma of neonatal rat DRG neurons and N18TG2 mouse neuroblastoma cells, possess a high proliferative activity with some characteristic features of sensory neurons (e.g., expression of substance P and the neurotrophin receptor, TrkA; formaldehyde-induced increases in intracellular Ca^2+^; and prostaglandin E2-induced neurite outgrowth via the second messenger, cAMP) [17,18]. Since the amount of protein obtained from primary cultured DRG neurons was insufficient for conducting Western blotting, we substituted them with ND7/23 cells. Although significant attention must be paid to the phenotypic differences between primary cultured DRG neurons and ND7/23 cells, the combined use of these two types of cells can be advantageous for the exploration of the molecular mechanisms underlying zonisamide’s neuroprotective properties [6,26].

The MAPKs are key elements of the signal transduction machinery and play a pivotal role in a wide range of reactions related to cell growth, differentiation, and death [27]. The involvement of MAPK signaling pathways in OHP-mediated neurotoxicity has been extensively investigated. For instance, OHP is known to evoke neuropathic pain and to upregulate the ERK phosphorylation in the DRG of rats [21]. Similarly, the OHP-induced neuropathy and the upregulation of ERK phosphorylation in the spinal cords of mice have been shown to be attenuated by PD0325901; a MEK inhibitor [28]. In another study, OHP has been shown to induce mechanical allodynia and enhanced phosphorylation of p38 MAPK, but not of ERK or JNK, in the spinal cords of mice [29]. In vitro, the OHP-induced cell death of embryonic rat DRG neurons is mediated by an upregulation of p38 MAPK and downregulation of JNK/SapK and ERK, and these changes were ameliorated by nerve growth factor (NGF) [12]. These findings are similar to those in our study, but we did not observe any significant changes concerning the JNK phosphorylation in OHP-treated ND7/23 cells. The differences in the OHP-induced alterations of the MEK/ERK or JNK/SapK signaling pathways among the aforementioned studies might stem from different experimental designs and cell sources. As both NGF and zonisamide have been shown to promote neurite outgrowth of DRG neurons via activation of the MEK/ERK signaling pathway [6,30], this pathway may play a role in the protective activity of these molecules against OHP-induced neurotoxicity at the cellular level. In contrast to the diverse outcomes regarding the MEK/ERK and the JNK/SapK pathways, there is fairly a general agreement regarding the pathological role of the p38 MAPK signaling pathway in OHP-induced neurotoxicity [29,31,32,33,34,35]. Consistent with those findings, we observed a significant upregulation of the p38 phosphorylation in OHP-treated ND7/23 cells and protective activity of the p38 inhibitor, SB239063, against the OHP-induced DRG neuronal death. As zonisamide attenuated the OHP-induced p38 phosphorylation and alleviated the OHP-induced neurotoxicity in a manner comparable to that of SB239063, it seems reasonable to suppose that the neuroprotective properties of zonisamide against OHP can be, at least partly, attributable to the suppression of the p38 MAPK pathway. Several molecules downstream of the p38 MAPK pathway have been suggested to be involved in the OHP-induced neurotoxicity (e.g., nuclear factor-κB (NF-κB), activating transcription factor 6 (ATF6), and tissue factor (TF)/hypoxia-inducible factor-1α (HIF-1α)) [33,34] and, therefore, our ongoing study focuses on the effects of the pretreatment with zonisamide on the expression of these molecules in the PNS exposed to OHP. The respective signals, such as p38/NF-κB, p38/ATF6, and p38/TF/HIF-1α, are suggested to induce neuronal apoptosis and inflammation, endoplasmic reticulum (ER) stress, and neural hypoxia, respectively. Therefore, it is important to specify the molecules and pathways downstream of p38 that are suppressed by zonisamide. Recent studies suggest that zonisamide prevents diabetes-related dementia and cardiomyopathy by inhibiting upregulated expression of ATF6 and other ER stress marker proteins [36,37].

Apart from the MAPK signaling pathways, we also focused on the PI3K/AKT pathway, which is known to play a role in the neurotrophic activity of zonisamide [6]. PI3K/AKT signaling has been suggested to play a pathogenetic role in OHP-induced neuropathic pain [38,39], whereas no studies so far have indicated its involvement in the actions of molecules reported to act as neuroprotective agents against OHP. In our study, zonisamide tended to enhance AKT phosphorylation in naïve ND7/23 cells but showed no effect on OHP-treated cells. In contrast, its alleviating effects on the OHP-induced DRG neuronal death were canceled by a co-incubation with the PI3K inhibitor, LY294002. The discrepant findings might result from the different cellular responses expected to exist between primary cultured DRG neurons and ND7/23 cells, or from the protocol followed for the performance of the Western blotting analysis (longer or shorter incubation of zonisamide and/or the fact that OHP might cause the changes in the phospho-AKT expression that are consistent with the survival assay). From these findings, it is difficult to make a clear statement regarding the role of the PI3K/AKT pathway in the actions of zonisamide; however, further discussion based on the co-culture experiments will be presented later.

In addition to the pretreatment of cells with zonisamide before their exposure to OHP, we performed a simultaneous application of zonisamide and OHP to DRG neurons but observed no significant restoring effects of zonisamide on the OHP-induced toxicity (Takaku et al., personal data). The reasons for the discrepant results in the efficacy of zonisamide between the pretreatment and the concurrent application remain unknown. When considering the findings of the signaling pathways discussed above and the efficacy of pre-administration of SB239063 toward neuroinflammation [40,41], the suppression of the p38 MAPK pathway ahead of the exposure to OHP might be critical for the zonisamide’s neuroprotective actions against OHP. It is also of interest to note that the pre-administration of several compounds has prevented neurological manifestations and mitochondrial dysfunction in mice exposed to OHP [42] and cisplatin [43]. These findings allow us to assume that the pretreatment of neuroprotective agents can be a promising therapeutic approach against chemotherapy-induced neuropathy.

In contrast to the protective effects on the DRG neurons and ND7/23 cells, zonisamide failed to alleviate the OHP-induced IFRS1 Schwann cell death. These findings are consistent with those of our previous study [6], in which zonisamide promoted the neurite outgrowth of DRG neurons but not the proliferation/survival or migration of IFRS1 Schwann cells. As the OHP-induced neuropathy can be classified as a “neuronopathy” and several reports have suggested that Schwann cells are less vulnerable to OHP than neurons [44,45], it seems reasonable to assume that zonisamide can alleviate OHP-induced neuropathy through its protective activities toward neurons rather than Schwann cells. In stark contrast to OHP, amiodarone hydrochloride (AMD), an anti-arrhythmic agent, has been shown to affect Schwann cells more severely than neurons [46]; a fact that may explain the “myelinopathy”-predominant features of the AMD-induced neuropathy [47]. However, it is important to note that longer OHP exposures resulted in a more prominent IFRS1 cell death in this study. Similarly, OHP has been reported to induce cell death and to decrease the myelin protein expression in primary cultured neonatal rat Schwann cells in a dose- and time-dependent manner [48]. Taking these findings into consideration, there is no denying that the OHP-induced toxicity against Schwann cells plays a role in the development of OHP-induced neuropathy.

Interestingly, zonisamide alleviated the OHP-induced neurite degeneration and detachment of IFRS1 cells from neurites in the DRG neuron–IFRS1 co-culture system. As zonisamide failed to exert protective activities toward the IFRS1 Schwann cells, its restoring effects on the co-culture could be attributed to the prevention of neuronal cell injury and subsequent axonal degeneration. Tsutsumi et al. [49] have observed a hypomyelination and reduced protein expression of the cleaved neuregulin (NRG)-1 type III and of the myelin basic protein (MBP) in sciatic nerves, as well as a reduced NRG-1 mRNA expression in the DRG of OHP-treated rats. The NRG-1 type III protein produced in DRG is transported to the axonal surface, where it is cleaved and bound to the ErbB receptor on Schwann cells to stimulate the PI3K/AKT signaling pathway and myelination [50]. Therefore, the OHP-induced toxicity against DRG neurons might secondarily induce hypomyelination or myelin sheath damage through the disruption of these signaling cascades. As zonisamide was able to prevent the reduction of AKT phosphorylation in the co-cultures exposed to OHP, its protective activities toward the co-cultured cells can be associated with the restoration of the PI3K/AKT signaling that plays a pivotal role in myelination [51]. Several downstream target molecules of the PI3K/AKT pathway (e.g., mammalian target of rapamycin, glycogen synthase 3β, cAMP response element-binding protein (CREB), Forkhead box O1/O3, and RhoA) are involved in the maintenance of neuronal survival and myelination [52]. Our current investigation is aimed at detecting the molecules relevant to the alleviating effects of zonisamide against OHP neurotoxicity. In a recent study, zonisamide ameliorated cognitive impairment by enhancing the activity and expression of CREB in the cortex and hippocampus of type 2 diabetic mice [36].

In summary, the pretreatment with zonisamide alleviated the OHP-induced DRG neuronal death, possibly via suppressing the p38 MAPK signaling pathway and by stimulating the MEK/ERK and the PI3K/AKT pathways (Figure 6). Although zonisamide exerted no protective effects on the death of OHP-treated IFRS1 Schwann cells, it prevented neurite degeneration and demyelination-like changes in the DRG neuron–IFRS1 co-cultures exposed to OHP. While these findings suggest the potential efficacy of the pretreatment with zonisamide against the neurotoxic adverse effects of OHP, the present study has several limitations. Firstly, in terms of the relevance of our study to OHP-induced neuropathy, in vivo analyses need to be conducted. We proceed with such experiments (Takaku et al., in preparation) referring to the previous studies that showed the efficacy of zonisamide toward neuropathic pain [7,8]. However, our DRG neuron-IFRS1 co-culture model appears to mimic axonal degeneration and regeneration better than single neuronal or Schwann cell culture models and can be a novel tool to evaluate the OHP-induced neuropathy and the alleviating effects of zonisamide against the lesion. In addition, the neuron-specific efficacy of zonisamide against the OHP neurotoxicity shown in this study might be useful for considering its clinical application toward chemotherapy-induced neuropathy. Secondly, as discussed above, further research is required to clarify which downstream molecules of the respective signaling pathways are more involved in the zonisamide’s alleviating effects against the OHP-induced neurotoxicity. Thirdly, it remains unclear whether the zonisamide’s anti-epileptic and anti-PD actions reported so far are also relevant to its anti-OHP activities. In particular, zonisamide has been shown to ameliorate epilepsy and PD through the inhibition of T-type Ca^2+^ channels [23,53]. As the OHP-induced upregulation of the function and the protein expression of the Ca^2+^ channels in the DRG neurons can be a cause of not only neuropathic pain but also apoptotic neuronal cell death [54], zonisamide’s protective activities against OHP might be, at least partly, attributable to the suppression of these channels. Zonisamide and other anti-epileptic drugs can block the repetitive firing of voltage-gated Na^+^ channels, leading to a reduction of T-type Ca^2+^ channel currents or by binding allosterically to GABA receptors [55]. It remains unclear if this action of zonisamide is associated with the signaling pathways involved in the OHP-induced neuropathy. However, other anti-convulsants, such as pregabalin and lacosamide, ameliorated the paclitaxel-induced peripheral neuropathy in rats through the inhibition of p38 MAPK phosphorylation [56]. As these drugs have also shown efficacy toward the OHP-neurotoxicity [57,58], the p38 MAPK signaling pathway might play a role in the OHP-induced activation of Na^+^ and Ca^2+^ channels. Fourthly, as OHP has been shown to suppress the growth of colorectal cancer, it is important to confirm that zonisamide does not interfere with the anti-cancer efficacy of OHP [59]. Our ongoing and future studies will elucidate these issues and will allow us to solidify zonisamide as a promising neuroprotective agent against the OHP-induced neurotoxicity. Because zonisamide has been approved as a safe and effective anti-epileptic and anti-PD agent, its repositioning for patients with OHP-induced neuropathy would be much more straightforward and timesaving than the clinical trials of unapproved drugs.

## 4. Materials and Methods

### 4.1. Animals

Female Wistar rats at 6–7 weeks of age were purchased from CLEA Japan, Inc. (Shizuoka, Japan). All rats were fed standard chow and water ad libitum and were housed in a temperature- and humidity-control room with a 12:12 h light–dark cycle. Three or less rats were kept in the same cage and all rats received humane care and handling following the ARRIVE guidelines, and all experiments were approved by the Institutional Review Board of the Tokyo Metropolitan Institute of Medical Science (institutional approval numbers: 20-007 and 22-007). Before the dissection, rats were anesthetized for euthanasia with 3% isoflurane (Abbott Japan, Tokyo, Japan) for 3 min.

### 4.2. Primary Culture of DRG Neurons and Cell Viability Assay

Dissociated cell cultures of DRG neurons were carried out as previously reported [22]. Briefly, 25-30 DRGs from the cervical to the lumbar levels were dissected from each animal and were dissociated with collagenase (CLS-3; Worthington Biochemicals, Freehold, NJ, USA) and trypsin (Sigma, St. Louis, MO, USA). These ganglia were subjected to density gradient centrifugation (5 min, 200× *g*) with 30% Percoll PLUS^TM^ (GE Healthcare Bio-Sciences Corp., Piscataway, NJ, USA) to eliminate the myelin sheath. This procedure resulted in a yield of more than 5 × 10^4^ neurons, along with a smaller number of non-neuronal cells. These neurons were suspended in Dulbecco’s Modified Eagle’s medium (DMEM)/Ham’s F12 (Thermo Fisher Scientific Inc., Waltham, MA, USA) supplemented with 10% fetal bovine serum (FBS; Thermo Fisher) and were seeded on poly-L-lysine (10 μg/mL; Sigma)-coated wells of 12-well culture plates (Thermo Fisher). Circles with a diameter of 0.9 cm were delineated by using black thin lines on the bottom of each well, and the neuronal cell density was adjusted to approximately 500–700 cells within a circle. After remaining in the serum-containing medium for 16 h, the neurons were incubated for 24 h in DMEM with serum-free medium supplement B27 (Thermo Fisher) in the presence or absence of 100 μM zonisamide (provided by Sumitomo Pharma Co., Ltd., Osaka, Japan). Subsequently, the neurons were maintained for 24 h in a fresh serum-free medium in the presence or absence of 75 μM of oxaliplatin (OHP; Wako, Osaka, Japan), and the viability of the neurons in each culture condition ((Control), (Zonisamide), (OHP), and (Zonisamide + OHP)) was assessed by trypan blue staining. Dead neurons were identified as lacking in phase-brightness under a phase-contrast light microscope (IMT-2; Olympus, Tokyo, Japan) and were confirmed by positive trypan blue staining. The representative phase-contrast micrographs of the respective experimental groups were taken by using a microscope digital camera system (DP22-CU; Olympus) and image analysis software (WinROOF2015; Mitani Corporation, Tokyo, Japan). The number of viable neurons in each circle on day 3 of the culture (24 h after their exposure to OHP) was expressed as a relative value wherein the original number on day 1 of the culture (before the pretreatment with zonisamide) was assumed to be 1.

To investigate the signaling pathways mediating the alleviating activity of zonisamide toward the OHP-induced toxicity, 25 μM of the MEK inhibitor U0126 (Calbiochem; EMD Chemicals, Inc., San Diego, CA, USA) and the PI3K inhibitor LY294002 (Cell Signaling Technology, Beverly, MA, USA) were co-administered along with 100 μM of zonisamide. The effects of 25 μM of the p38 MAPK inhibitor SB239063 (Calbiochem) against the OHP-induced toxicity were also examined.

### 4.3. Culture of ND7/23 Sensory Neuron-Like Cells and IFRS1 Schwann Cells and Cell Viability Assay

The ND7/23 cells were kindly provided by Prof. Atsufumi Kawabata and Dr. Fumiko Sekiguchi of Kindai University, Osaka, Japan. IFRS1 Schwann cells were established in our laboratory [19]. The ND7/23 cells at a passage of 15–20, as well as IFRS1 cells at a passage of 30–40, were maintained in DMEM (Thermo Fisher) supplemented with 5% FBS, and the toxicity of OHP toward these cells was evaluated by using the Cell Titer 96^®^ AQueous One Solution Cell Proliferation Assay kit (Promega, Madison, WI, USA) as previously described [26]. Briefly, the cells were seeded in each well of 96-well culture plates (Thermo Fisher) at an approximate density of 3 × 10^4^/cm^2^ and incubated for 24 h in DMEM containing 1% FBS in the presence or absence of 100 μM zonisamide and were then maintained for 24–48 h in fresh DMEM containing 1% FBS in the presence or absence of 75 μM OHP. After rinsing with serum-free DMEM, the cells were incubated for 1 h at 37 °C in DMEM along with the Cell Titer 96^®^ Aqueous One Solution Reagent, and the absorbance was determined at 490 nm with a microplate reader (Varioskan Flash; Thermo Fischer). 

### 4.4. Co-Culture of DRG Neurons and IFRS1 Schwann Cells

Co-culturing of DRG neurons and IFRS1 cells was conducted as previously described [22], with slight modifications (Figure 5a). Briefly, DRG neurons were seeded on type I collagen-coated chamber slides (Matsunami Glass Ind., LTD, Osaka, Japan) and Aclar fluorocarbon coverslips (Nissin EM Co., Tokyo, Japan) at an approximate density of 2 × 10^3^/cm^2^, and were maintained for 7 days in DMEM/F12 with N2 supplement (Thermo Fisher), 10 ng/mL NGF (R&D Systems, Inc., Minneapolis, MN, USA), 10 ng/mL glial cell line-derived neurotrophic factor (R&D Systems), and 10 ng/mL ciliary neurotrophic factor (Peprotech, Rocky Hill, NJ, USA). After confirming the neurite elongation from the neuronal cell bodies under a phase-contrast microscope, IFRS1 cells are added to the neurons at an approximate density of 2 × 10^4^/cm^2^; the co-cultured cells were incubated for 2 days in DMEM/F12 containing 5% FBS, and subsequently maintained for 14 days in DMEM/F12/B27 with 50 μg/mL ascorbic acid (Wako) to induce myelination. The cells were then incubated for 2 days in the serum-free medium in the presence or absence of 100 μM zonisamide and for 7 days in the presence of 10 μM OHP.

### 4.5. Western Blotting Analysis

Western blotting analysis was conducted as previously described [26], with slight modifications. Briefly, ND7/23 cells were incubated for 1 h in DMEM containing 1% FBS in the presence or absence of 100 μM zonisamide, then maintained for 1 h in fresh DMEM containing 1% FBS in the presence or absence of 75 μM OHP. Protein was extracted from the cells under each culture condition by using a 1× sodium dodecyl sulfate (SDS) sample buffer. The cell extracts were resolved using SDS-polyacrylamide gel electrophoresis (SDS-PAGE) in 5–20% SDS-PAGE gel (Fujifilm, Tokyo, Japan), and were transferred onto a polyvinylidene fluoride membrane by using an electroblotter (Nihon Eido Co., Ltd., Tokyo, Japan). The membrane was incubated in phosphate-buffered saline (PBS) with 0.1% Tween 20 (including 5% skimmed milk or 3% bovine serum albumin) for 1 h at room temperature, and then overnight at 4 °C with the following antibodies: (i) anti-p38 MAPK antibody (1:1000; Cell Signaling (8690)), (ii) anti-phospho-p38 MAPK antibody (Thr180/Tyr182) (1:1000; Cell Signaling (4511)), (iii) anti-JNK antibody (1:1000; Cell Signaling (9252)), (iv) anti-phospho-JNK antibody (Thr183/Tyr185) (1:1000; Cell Signaling (4668)), (v) rabbit anti-ERK1/2 antibody (1:1000; Cell Signaling (4695)), (vi) rabbit anti-phospho-ERK1/2 antibody (Thr202/Tyr204) (1:1000; Cell Signaling (4370)), (vii) rabbit anti-AKT polyclonal antibody (1:1000; Cell Signaling (9272)), and (viii) rabbit anti-phospho-AKT polyclonal antibody (Thr308) (1:1000; Cell Signaling (13038)). After rinsing with PBS, the membrane was incubated in a solution of horse radish peroxidase-conjugated anti-rabbit IgG antibody or anti-mouse IgG antibody (1:2000; MBL Corp., Ltd., Nagoya, Japan) for 1 h. After rinsing, immunocomplexes on the membrane were visualized with enhanced chemiluminescence by using a Western Lightning^TM^ Ultra (PerkinElmer, Inc., Waltham, MA, USA).

Protein extraction from the co-cultured cells after 7 days incubation with 10 μM OHP was conducted in a similar manner to ND7/23 cells, and rabbit anti-AKT and anti-phospho-AKT polyclonal antibodies were used for Western blotting analysis.

### 4.6. Statistical Analysis

All data are presented as means followed by SD, and the number of experiments is indicated in the figure legends. Parametric comparisons between experimental groups were performed by one-way analysis of variance (ANOVA); when ANOVA showed a significant difference between groups (*p* < 0.05), a Tukey–Kramer test was used to identify which exact group differences accounted for the significant *p*-value.

## Figures and Tables

**Figure 1 ijms-23-09983-f001:**
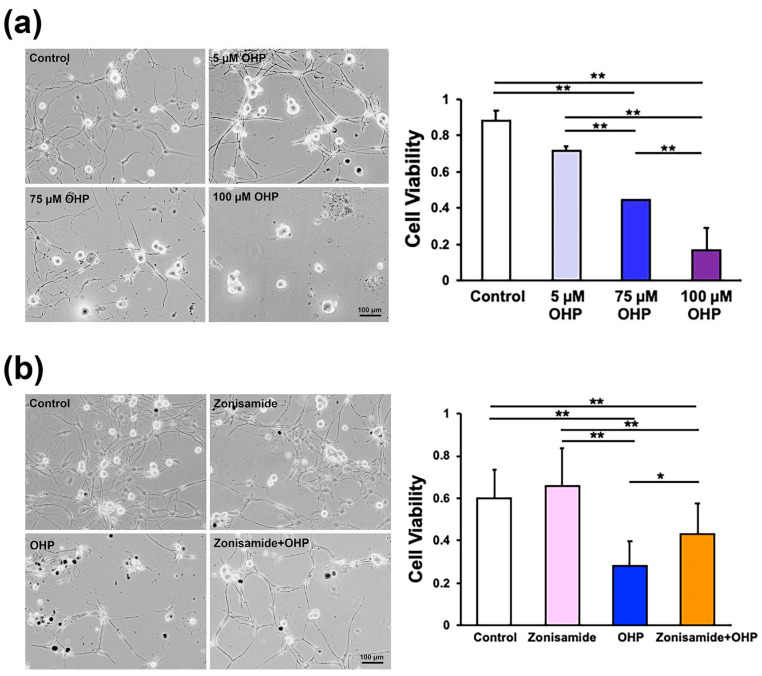
Pretreatment with zonisamide alleviates OHP-induced toxicity against primary cultured adult rat DRG neurons: (**a**) OHP induces DRG neuronal death in a dose-dependent manner; the representative phase-contrast micrographs of DRG neurons (left) and the average cell viability ratios at 24 h after the exposure to the different concentrations of OHP (right). (**b**) Preincubation of 100 μM zonisamide (for 24 h) diminishes the cell death due to exposure to 75 μM of OHP for 24 h; representative phase-contrast micrographs of DRG neurons (left) and the average cell viability ratios at 24 h after the exposure to the respective culture conditions (right). Data are presented as the mean ± standard deviation (*SD*) of (**a**) 6 and (**b**) 16 experiments, respectively; **: *p* < 0.01; *: *p* < 0.05 (as defined by Tukey–Kramer).

**Figure 2 ijms-23-09983-f002:**
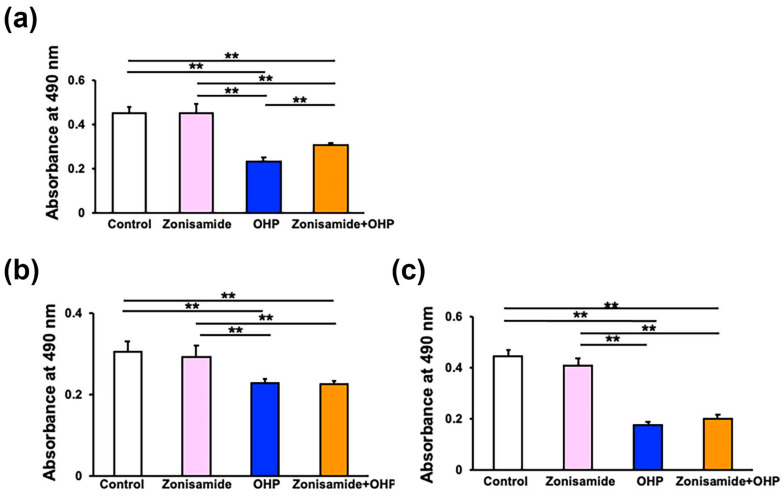
Pretreatment with zonisamide alleviates the OHP-induced toxicity against ND7/23 cells, but not against IFRS1 cells; results refer to an MTS assay: (**a**) Preincubation of zonisamide (100 μM) for 6 h mitigates the ND7/23 cell death due to exposure to OHP (75 μM) for 24 h. (**b**,**c**) Preincubation of zonisamide (100 μM) for 6 h fails to mitigate the cell death caused as a result of exposure to OHP (75 μM) for (**b**) 24 h or (**c**) 48 h. Data are presented as the mean ± *SD* of (**a**) 16, (**b**) 12, and (**c**) 30 experiments, respectively; **: *p* < 0.01 (as defined by Tukey–Kramer).

**Figure 3 ijms-23-09983-f003:**
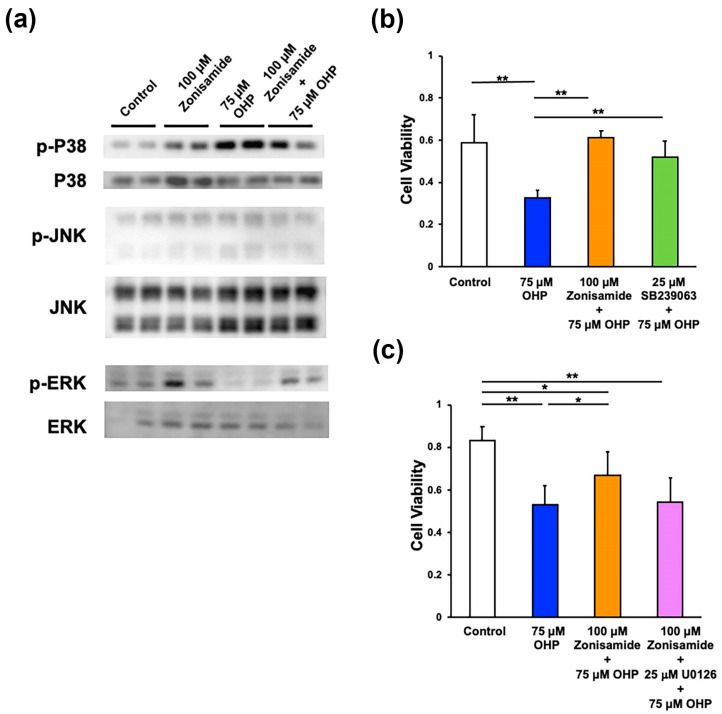
Involvement of MAPK signaling pathways in the protective activity of zonisamide against OHP-induced neurotoxicity: (**a**) Preincubation with 100 μM of zonisamide for 1 h attenuates the upregulated expression of the phosphorylated p38 MAPK (p-P38) as well as the downregulated expression of the phosphorylated ERK (p-ERK) in ND7/23 cells exposed to OHP (75 μM) for 1 h; results referring to Western blotting. The signals for p-P38/P38 and p-JNK/JNK were detected by using the same membrane. (**b**) Preincubation with 100 μM of zonisamide or with 25 μM of SB239063 (a p38 MAPK inhibitor) for 24 h diminishes the cell death as a result of exposure to 75 μM of OHP for 24 h. (**c**) Co-incubation with 25 μM of U0126 (a MEK inhibitor) weakens the alleviating effects of zonisamide (100 μM) against the OHP-induced (75 μM) cell death. Data are presented as the mean ± *SD* of (**b**) 6 and (**c**) 8 experiments, respectively; **: *p* < 0.01; *: *p* < 0.05 (as defined by Tukey–Kramer).

**Figure 4 ijms-23-09983-f004:**
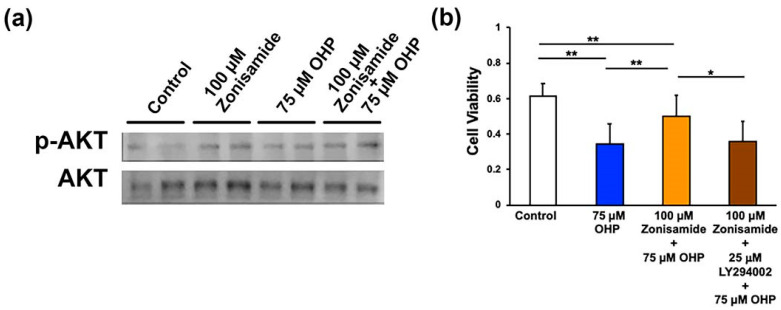
Involvement of the PI3K/AKT signaling pathway in the protective activity of zonisamide against OHP-induced neurotoxicity: (**a**) Preincubation with 100 μM zonisamide for 1 h tended to upregulate the expression of phosphorylated AKT (p-AKT) in naïve ND7/23 cells, but not in the cells incubated with 75 μM of OHP for 1 h; results referring to Western blotting. (**b**) Co-incubation with 25 μM of LY294002 (a PI3K inhibitor) weakens the alleviating effects of zonisamide (100 μM) against the OHP-induced (75 μM) cell death. Data are presented as the mean ± *SD* of 11 experiments; **: *p* < 0.01; *: *p* < 0.05 (as defined by Tukey–Kramer).

**Figure 5 ijms-23-09983-f005:**
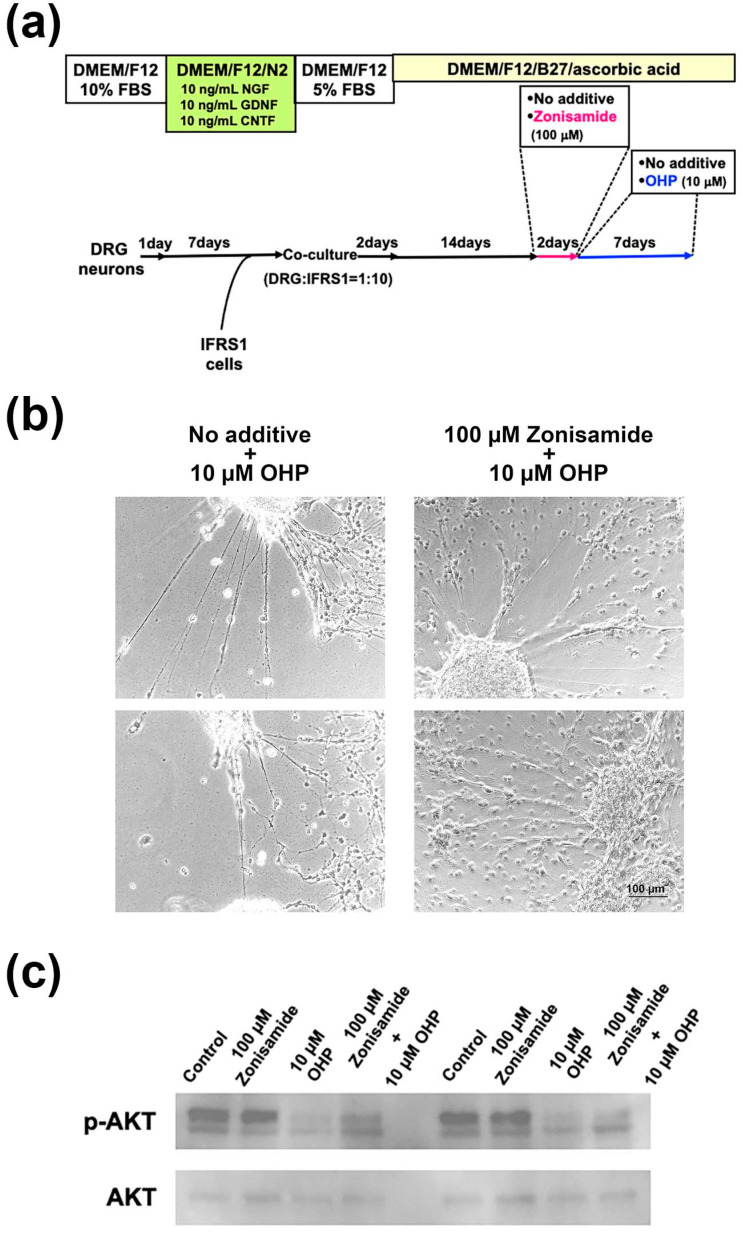
Zonisamide pretreatment alleviates the OHP-induced neurite degeneration as well as the demyelination-like changes in DRG neuron–IFRS1 co-cultures: (**a**) Schematic representation of the co-culture-generating procedure and the applications of zonisamide and OHP. (**b**) Representative phase-contrast micrographs of the co-cultures at 7 days after the OHP treatment in the presence (right) or absence (left) of a 2-day pretreatment with zonisamide. The pictures in the respective culture conditions were taken from the different co-cultures. (**c**) Preincubation with zonisamide (100 μM) for 2 days attenuates the downregulated expression of phosphorylated AKT (p-AKT) in co-cultured cells exposed to OHP (10 μM) for 7 days; results refer to Western blotting.

**Figure 6 ijms-23-09983-f006:**
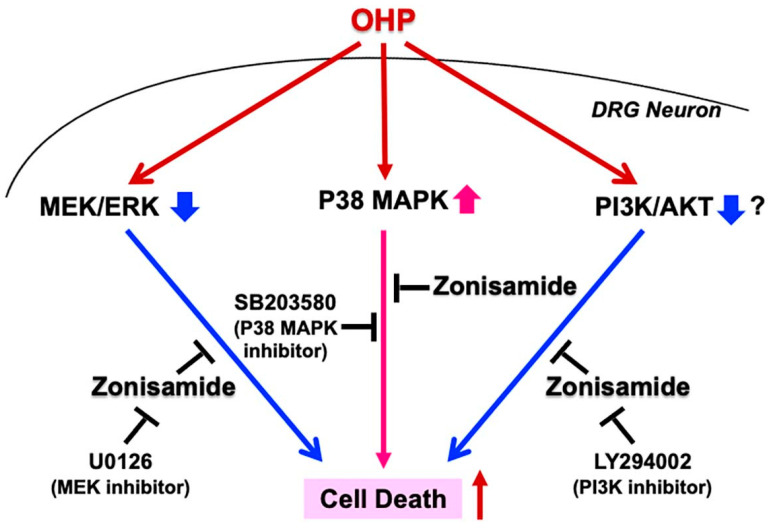
A schematic diagram showing the signaling pathways involved in the alleviating effects of zonisamide against OHP-induced DRG neuronal cell death based on the present study.

## Data Availability

The data presented in this study are available upon request from the corresponding author.

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
