# Peer review of "Pretreatment with Zonisamide Mitigates Oxaliplatin-Induced Toxicity in Rat DRG Neurons and DRG Neuron–Schwann Cell Co-Cultures"

_ijms, 2022, doi:10.3390/ijms23179983_

Round 1

Author Response

Response to Reviewer 1 Comments

General comments: I have read the manuscript “Pretreatment with zonisamide mitigates oxaliplatin-induced toxicity in rat DRG neurons and DRG neuron–Schwann cell co- cultures” and found it novel and suitable for publication. The manuscript is well written, organized and there is connection between sentences.

Response: We thank the reviewer for giving a high evaluation on our manuscript. We believe its quality has been further improved by the revision based on the reviewers’ helpful comments.

Point 1: Abstract

Add 1-3 line for future recommendation

Response 1: We added the following sentence to the end of the abstract (Lines 23-24 in the revised manuscript).

Future studies will allow us to solidify zonisamide as a promising remedy against the neurotoxic adverse effects of OHP.

Point 2: Introduction

 Line 31 “as compared with” should be replaced by “as compared to”

Response 2: As the reviewer suggested, we changed the word (Line 31 in the revised manuscript).

Point 3:  Results, Discussion and Methodology is well written and fully supported using appropriate references. Please read the manuscript and deal with grammatical errors where necessary. Since mice are used no ethical concern has been discussed in this study.

Response 3: We thank the reviewer for the concern of the whole manuscript. Although the original manuscript was checked by a native English speaker prior to the submission, we carefully read the revised manuscript and corrected a typo (Line 375 in the revised manuscript). Ethical issues regarding animal cares have been documented in more detail in Materials and Methods section (Lines 426-428 in the revised manuscript).

Reviewer 2 Report

The authors investigated the attenuated effects of zonisamide on oxaliplatin-induced toxicity using primary rat DRG neurons and DRG neuron-Schwann cell co-culture. They reported the neuroprotective effect of pretreatment with zonisamide in OHP-induced peripheral sensory neuropathy, which is related to the activation of the MEK/ERK and PI3K/AKT signaling and suppression of the p38 MAPK signaling in DRG neurons. This topic is interesting. However, the methods, results and conclusions are insufficient and the clinical implications need to be explained to the reader in more detail.

Some concerns are listed as follows,

Although various signal pathways are involved in zonisamide’s alleviating effects on the OHP-induced DRG neuronal death, the data presented in this study did not fully reflect the specificity and significance of the signal pathways. It should be discussed in detail.

Zonisamide can block repetitive firing of voltage-gated sodium channels, leading to a reduction of T-type calcium channel currents or by binding allosterically to GABA receptors. Dose this action of zonisamide relates to the signaling pathways involved in the OHP-induced peripheral sensory neuropathy. More evidence and explanation should be provided.

Although primary cells were used in this study, the real effect of zonisamide in vivo is still unclear, leading insufficient potential in clinical implication. Animal experiments are recommended.

Author Response

Response to Reviewer 2 Comments

General comments: The authors investigated the attenuated effects of zonisamide on oxaliplatin-induced toxicity using primary rat DRG neurons and DRG neuron-Schwann cell co-culture. They reported the neuroprotective effect of pretreatment with zonisamide in OHP-induced peripheral sensory neuropathy, which is related to the activation of the MEK/ERK and PI3K/AKT signaling and suppression of the p38 MAPK signaling in DRG neurons. This topic is interesting.

Response: We thank the reviewer for reading our manuscript. We believe its quality has been improved by the revision based on the reviewers’ helpful comments.

Point 1: However, the methods, results and conclusions are insufficient, and the clinical implications need to be explained to the reader in more detail.

Response 1: As the reviewer suggested, we described more detailed information regarding the methods (Pages 11-13), results (Pages 2-7) and conclusions (Pages 10-11 in the revised manuscript). We addressed the ongoing and future studies which would be essential for the repositioning of zonisamide toward OHP-induced neuropathy (Lines 270-291, 363-369, 378-386, 395-405, and 416-419) in the revised manuscript). We provided additional data with 1970C3 mouse Schwann cells [Niimi et al., J Neurochem 2018] (Figure S1 in the revised manuscript) to suggest that zonisamide exerts no protective activity against OHP-treated Schwann cells regardless of cell lines (Lines 103-106 in the revised manuscript).

Point 2: Although various signal pathways are involved in zonisamide’s alleviating effects on the OHP-induced DRG neuronal death, the data presented in this study did not fully reflect the specificity and significance of the signal pathways. It should be discussed in detail.

Response 2: We thank the reviewer for the comments on the signaling pathways.

  • For readers’ convenience, a schematic diagram showing the signaling pathways involved in the alleviating effects of zonisamide against OHP-induced DRG neuronal cell death based on the present study is provided (Figure 6 in the revised manuscript).
  • As described in the text (Pages 8-9), there is a divergence among previous studies in the effects of OHP on JNK/SapK, MEK/ERK and PI3K/AKT signaling pathways. In contrast, there is fairly a general agreement regarding the pathological role of p38 MAPK signaling pathway in OHP-induced neurotoxicity. We focused more on the significance of this pathway in relation to the neuroprotective activity of zonisamide. The respective signals, such as p38/NF-kB, p38/ATF6, and p38/TF/HIF-1a, are suggested to induce neuronal apoptosis and inflammation, endoplasmic reticulum (ER) stress, and neural hypoxia, respectively. Therefore, it is important to specify the molecules and pathways downstream of p38 that are suppressed by zonisamide. Recent studies suggest that zonisamide prevents diabetic dementia and cardiomyopathy by inhibiting upregulated expression of ATF6 and other ER stress marker proteins [He et al., Front Aging Neurosci 2020; Tian et al., Acta Pharmacol Sin 2021] (Lines 270-291 in the revised manuscript).
  • In addition, we fully delineated the downstream target molecules of the PI3K/AKT pathway. Our current investigation is aimed to detect the molecules relevant to the alleviating effects of zonisamide against OHP neurotoxicity. In a recent study, zonisamide ameliorated cognitive impairment by enhancing the activity and expression of CREB in the cortex and hippocampus of type 2 diabetic mice [He et al., Front Aging Neurosci 2020] (Lines 363-369 in the revised manuscript).

Point 3:  Zonisamide can block repetitive firing of voltage-gated sodium channels, leading to a reduction of T-type calcium channel currents or by binding allosterically to GABA receptors. Dose this action of zonisamide relates to the signaling pathways involved in the OHP-induced peripheral sensory neuropathy. More evidence and explanation should be provided.

Response 3: We thank the reviewer for the important suggestion, and we cited a paper describing the actions of zonisamide and other anti-epileptic drugs [Landmark, Med Sci Monit 2007]. However, no paper has addressed the association of this action of zonisamide with the signaling pathways involved in the OHP-induced neuropathy. Instead, other anti-epileptic drugs, such as pregabalin and lacosamide, ameliorated the paclitaxel-induced peripheral neuropathy in rats through the inhibition of p38 MAPK phosphorylation [Al-Massri et al., Neurochem Int 2018]. As these drugs have also shown the efficacy toward the OHP-neurotoxicity [Di Cesare Mannelli et al., Cancer Chemother Pharmacol 2017; Argyriou et al., J Peripher Nerv Syst 2020], p38 MAPK signaling pathway might play a role in the OHP-induced activation of Na+ and Ca2+ channels (Lines 395-403 in the revised manuscript).

Point 4:  Although primary cells were used in this study, the real effect of zonisamide in vivo is still unclear, leading insufficient potential in clinical implication. Animal experiments are recommended.

Response 4: We agree to the authors’ comments and proceed with the in vivo experiments referring to the previous studies that showed the efficacy of zonisamide toward neuropathic pain [Bektas et al., Life Sci 2014; Koshimizu et al., Life Sci 2020]. However, our DRG neuron-IFRS1 co-culture model appears to mimic axonal degeneration and regeneration better than single neuronal or Schwann cell culture models and can be a novel tool to evaluate the OHP-induced neuropathy and the alleviating effects of zonisamide against the lesion. In addition, the neuron-specific efficacy of zonisamide against the OHP neurotoxicity shown in this study might be useful for considering its clinical application toward chemotherapy-induced neuropathy (Lines 378-386 in the revised manuscript).

Round 2

Reviewer 2 Report

No further comment.